# Chaos Synchronization for Hyperchaotic Lorenz-Type System via Fuzzy-Based Sliding-Mode Observer

**Corina Plata [1], Pablo J. Prieto [2], Ramon Ramirez-Villalobos [1],* and Luis N. Coria [1]**

[1] Tecnológico Nacional de México/IT de Tijuana, Calz. del Tecnológico S/N, Tomás Aquino, 22414 Tijuana, BC, Mexico; corina.plata@tectijuana.edu.mx (C.P.), luis.coria@tectijuana.edu.mx (L.N.C.)

[2] Cetys Universidad, Calz Cetys 813, Lago Sur, 22210 Tijuana, BC, Mexico; pablojprieto@ieee.org

* Correspondence: ramon.ramirez@tectijuana.edu.mx

**Abstract:** Hyperchaotic systems have applications in multiple areas of science and engineering. The study and development of these type of systems helps to solve diverse problems related to encryption and decryption of information. In order to solve the chaos synchronization problem for a hyperchaotic Lorenz-type system, we propose an observer based synchronization under a master-slave configuration. The proposed methodology consists of designing a sliding-mode observer (SMO) for the hyperchaotic system. In contrast, this type of methodology exhibits high-frequency oscillations, commonly known as chattering. To solve this problem, a fuzzy-based SMO system was designed. Numerical simulations illustrate the effectiveness of the synchronization between the hyperchaotic system obtained and the proposed observer.

**Keywords:** hyperchaotic system; fuzzy-based sliding-mode observer; synchronization

## 1. Introduction

Chaotic systems are nonlinear aperiodic oscillators with high sensitivity to initial conditions. Due to above, in recent years, the study of chaotic systems has increased because of their different applications in various areas of engineering [1]. In 1963, Edward N. Lorenz developed the first chaotic system of third order Ordinary Differential Equations (ODE). He realized that any variation in the initial conditions affects the final conditions [2]. Later, in 1979, Otto Rössler proposed the first hyperchaotic system. This system consists of a system of ordinary differential equations of four dimensions [3]. Unlike chaotic systems, hyperchaotic systems have a more complex behavior, i.e., their dynamics are expanded in more than one direction, giving rise to a more complex attractor [4].

Due to aforementioned, hyperchaotic systems have potential applications in different branches of science and engineering, for example: mobile robotics [5], secure communications systems [6], and encryption in biometric systems [7], among others. An interesting challenge for the scientific community is the chaos synchronization of hyperchaotic systems. In this context, the dynamic behavior of two systems must converge on the same unique chaotic behavior. These two systems can be coupled unidirectionally, also so-called *master-slave configuration*, i.e., the autonomous system with hyperchaotic dynamics is called *master* and the another system, which is forced to follow the hyperchaotic behavior by coupled inputs, called *slave* [1].

Chaos synchronization problem can be solved using various approaches, such as linear state error feedback control, observer-based synchronization [8,9], and adaptive synchronization [10], among others. Within this framework, the main idea of observer based synchronization methods is to implement an observer system as a *slave*. Sliding-mode observers (SMO) have been widely used in several applications [11], including chaos synchronization for continuous and sampled systems [12–16]. In contrast, one crucial problem of the SMO is the high-frequency oscillations,

also called *chattering*, caused by the discontinuous implementation of the *set-valued function*, latency from the measurement to the actuation and the sampling time delay [17]. Commonly, in order to suppress the high-frequency chattering, a low-pass filter structure is added [18]. However, the system exhibits a delayed response brought by the filter [19]. Another method of chattering suppression is the approximation of discontinuous control by using the boundary layer control [20,21]. In this method, the control is discontinuous outside boundary layer and continuous inside. This implies that the performance is similar to sliding-mode control (SMC) when the dynamic system is outside the boundary layer. Therefore, when the dynamic is inside the boundary layer the system can exhibit overshoot response. In order to overcome the chattering problem, fuzzy-based sliding-mode control (FSMC) has been proposed as an alternative to reduce chattering [22]. Significant researches has been done in this field. For instance, in Reference [22,23], the set-valued function is replaced by a fuzzy inference. In this alternative, the sliding-mode variable represents the input for fuzzy inference system; as a result of that, chattering attenuation is achieved.

In this paper, we propose a observer based synchronization under a master-slave configuration. Such scheme, lie in a fuzzy-based sliding-mode observer (FSMO) design for a hyperchaotic Lorenz-type system reported in Reference [24]. Firstly, the proposed methodology consists in designing a SMO for the hyperchaotic system by means of Lyapunov analysis. Finally, in order to avoid the high-frequency oscillations caused by SMO, a fuzzy inference system is designed to replace the set-valued function. In this case, the fuzzy inference system designed attenuate the high-frequency oscillations in the estimated variables.

The rest of this paper is organized as follows. Section 2 introduces the nonlinear mathematical model of the hyperchaotic Lorenz-type system and provides the background material necessary to understand the observer design. Section 3 introduces SMO and FSMO design to realize chaos synchronization. Section 4 provides emulations numerical simulations in order to show the effectiveness of the proposed approach. Finally, Section 5 presents the concluding remarks.

*Notations.* In this paper, the notation $\lambda_{\min}\{A\}$ ($\lambda_{\max}\{A\}$) is the minimum (maximum) eigenvalue of a matrix $A \in \mathbb{R}^{n \times n}$. We denote by $|x|$ the 1-norm of vector $x \in \mathbb{R}^n$, whereas $\|A\|$ denotes the induced norm of a matrix $A$.

## 2. Mathematical Model

Recently, Layek and Pati have introduced a four-dimensional mathematical model which describes a hyperchaotic behavior in magnetoconvection of couple-stress fluid system by imposing a vertical magnetic field is studied. This model is given by the following equations [24]:

$$\dot{x}_1 = \sigma(x_2 - cx_1 - qx_4), \tag{1a}$$

$$\dot{x}_2 = rx_1 - x_2 - x_1x_3, \tag{1b}$$

$$\dot{x}_3 = x_1x_2 - bx_3, \tag{1c}$$

$$\dot{x}_4 = \zeta(\alpha x_1 - x_4), \tag{1d}$$

where $b$, $c$, $r$, $q$, $\alpha$, $\sigma$, and $\zeta$ are positive constants. System (1) exhibits various attractors (e.g., periodic and quasi-periodic attractors, chaotic and hyperchaotic attractors) depending on the value of parameter $r$. For example, system (1) exhibits a hyperchaotic behavior (see Figure 1) for the following parameter values:

$$b = 8/3, \ c = 1 + (159/200)\pi^2, \ r = 600, \ q = 1573, \alpha = 4/(9\pi^2), \ \sigma = 10, \ \zeta = 0.1. \tag{2}$$

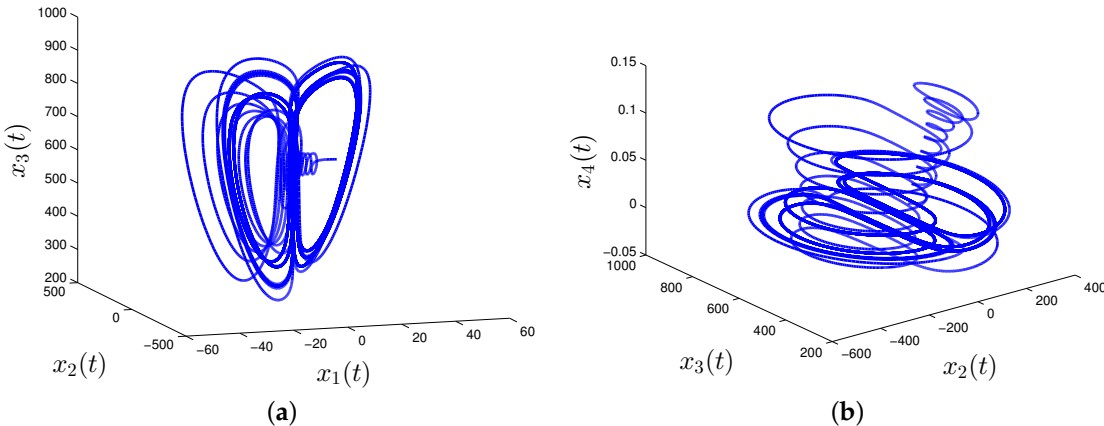

**Figure 1.** Chaotic attractor of system (1): (**a**) Phase space $(x_1, x_2, x_3)$. (**b**) Phase space $(x_2, x_3, x_4)$.

The following is henceforth assumed:

**Assumption 1.** *The nonlinear terms in* (1) *are bounded, with a priori known upper bounds; that is, there exists a positive constant* $F_1^+$ *and* $F_2^+$ *such that*

$$||x_1 x_3|| \leq F_1^+, \quad ||x_1 x_2|| \leq F_2^+. \tag{3}$$

This assumption is made for technical reasons. Assumption 1 allows us to establish sufficient conditions to guarantee convergence of estimations in the proposed observer design.

## 3. Observer Design

In this section, we design a sliding-mode observer considering $y = [x_2, x_3]^T$ as the measurable output of system (1). The proposed SMO has the following form:

$$\dot{\hat{x}}_1 = \sigma(\hat{x}_2 - c\hat{x}_1 - q\hat{x}_4), \tag{4a}$$

$$\dot{\hat{x}}_2 = r\hat{x}_1 - \hat{x}_2 + L_1 \operatorname{sgn}(x_2 - \hat{x}_2), \tag{4b}$$

$$\dot{\hat{x}}_3 = -b\hat{x}_3 + L_2 \operatorname{sgn}(x_3 - \hat{x}_3), \tag{4c}$$

$$\dot{\hat{x}}_4 = \zeta(\alpha\hat{x}_1 - \hat{x}_4), \tag{4d}$$

where $\hat{x}$ denotes the estimate of $x$, $L_1$ and $L_2$ are the observer gains, and $\operatorname{sgn}(\cdot)$ is the *set-valued function* defined as follows:

$$\operatorname{sgn}(a) := \begin{cases} a/|a|, & \text{if } a \neq 0 \\ [-1, 1], & \text{if } a = 0 \end{cases}, \tag{5}$$

where $a$ is a real value.

The observation error is defined as $e_i = x_i - \hat{x}_i$; $i = 1, 2, 3, 4$. Therefore, from system (1) and (4), the observer error dynamics is described as follows:

$$\dot{e}_1 = \sigma(e_2 - ce_1 - qe_4), \tag{6a}$$

$$\dot{e}_2 = re_1 - e_2 - x_1 x_3 - L_1 \operatorname{sgn}(e_2), \tag{6b}$$

$$\dot{e}_3 = x_1 x_2 - be_3 - L_2 \operatorname{sgn}(e_3), \tag{6c}$$

$$\dot{e}_4 = \zeta(\alpha e_1 - e_4). \tag{6d}$$

*3.1. Stability Analysis*

Consider the following Lyapunov function candidate for the error dynamics in system (6):

$$V(e) = \frac{\alpha}{\sigma}e_1^2 + e_2^2 + e_3^2 + \frac{q}{\zeta}e_4^2. \tag{7}$$

The time derivative of $V(e)$ along the trajectories of the observer error dynamics in system (6) yields:

$$\dot{V}(e) = -e^T Q e - 2(L_1 \operatorname{sgn}(e_2) - x_1 x_3)e_2 - 2(L_2 \operatorname{sgn}(e_3) - x_1 x_2)e_3 \tag{8}$$

with

$$Q := \begin{bmatrix} 2c\alpha & -(\alpha+r) & 0 & 0 \\ -(\alpha+r) & 2 & 0 & 0 \\ 0 & 0 & 2b & 0 \\ 0 & 0 & 0 & 2q \end{bmatrix}.$$

From Equation (8), we can conclude that the matrix $Q$ is positive definite if $4c\alpha > (r+\alpha)^2$ is fulfilled. Now, under the Assumption 1, Equation (8) is reduced to

$$\dot{V}(e) \le -\lambda_{\min}\{Q\}\|e\|^2 - 2(L_1 - F_1^+)|e_2| - 2(L_2 - F_2^+)|e_3|. \tag{9}$$

Selecting the observer gains $L_1$ and $L_2$ such that

$$L_1 > F_1^+ \tag{10a}$$
$$L_2 > F_2^+, \tag{10b}$$

and defining $\eta_1 := 2(L_1 - F_1^+)$ and $\eta_2 := 2(L_2 - F_2^+)$, Equation (9) can be rewritten as

$$\dot{V}(e) \le -\lambda_{\min}\{Q\}\|e\|^2 - \eta_1|e_2| - \eta_2|e_3| < 0. \tag{11}$$

Therefore, we can conclude that Equation (11) is a negative definite function; thus, the estimation errors converge to zero.

*3.2. Fuzzy-Based Sliding-Mode Observer*

The main problem of the SMO, as in system (6), is that the discontinuous implementation of sgn **function** in a real hardware exhibits *chattering*. Under this scenario, fuzzy inference system represents an alternative in order to avoid the chattering at the output of the observer, also as in system (6) [23,25]. In this case, a fuzzy inference system is designed to replace the set-valued function $\operatorname{sgn}(\cdot)$ so that the output $y$ is relaxed.

Let us consider the FSMO:

$$\dot{e}_1 = \sigma(e_2 - ce_1 - Qe_4), \tag{12a}$$
$$\dot{e}_2 = re_1 - e_2 - x_1 x_3 - L_1 \psi(e_2), \tag{12b}$$
$$\dot{e}_3 = x_1 x_2 - be_3 - L_2 \psi(e_3), \tag{12c}$$
$$\dot{e}_4 = \zeta(\alpha e_1 - e_4). \tag{12d}$$

Here, $\psi(\cdot) \in [-1,1] \subseteq \mathbb{R}$ is the fuzzy inference system, which is represented by the following *if-then* rule:

$$R_i : \text{If } e_i \text{ is } M_i \text{ then } \psi_i \text{ is } U_i, \tag{13}$$

where the observation error $e_i$ is the fuzzy variable at the input, and $U_i$ is the fired crisp value of the output $\psi_i$ regarding the value of $e_i$. The whole universe of discourse for both $e_i$ and $\psi_i$ is represented by five fuzzy sets: NB (Negative bigger), NS (Negative small), ZR (Zero), PS (Positive small), and PB (Positive bigger):

1. The states NB and PB represent the situation where error $e_i$ is too far from the origin in negative side and positive side, respectively.
2. The states NS and PS represent the situation where error $e_i$ is in the negative side and positive side, respectively.
3. The state ZR represents the situation when error $e_i$ is around the origin.

The fuzzy system is represented in Figure 2, where triangular membership functions for the input variable $e_i$ are defined as follows:

$$M_i := \begin{cases} \frac{e_i - \Phi_{i-1}}{\Phi_i - \Phi_{i-1}} & \text{if } \Phi_{i-1} \leq e_i < \Phi_i \\ \frac{e_i - \Phi_{i+1}}{\Phi_i - \Phi_{i+1}} & \text{if } \Phi_i \leq e_i < \Phi_{i+1} \\ 0 & \text{elsewhere.} \end{cases} \tag{14}$$

The whole universe of discourse of $e_i$ is partitioned in five triangular membership functions. Here, $e_i \in \{-\Phi_2, -\Phi_1, \Phi_0, \Phi_1, \Phi_2\}$ where $\Phi_{-i} = \Phi_i$ and $M_0(0) = 0$. The membership functions for the fuzzy sets of the output $\psi \in \{-U_2, -U_1, U_0, U_1, U_2\}$ are singleton-type functions where $U_i = -U_{-i}$ and $U_0 = 0$.

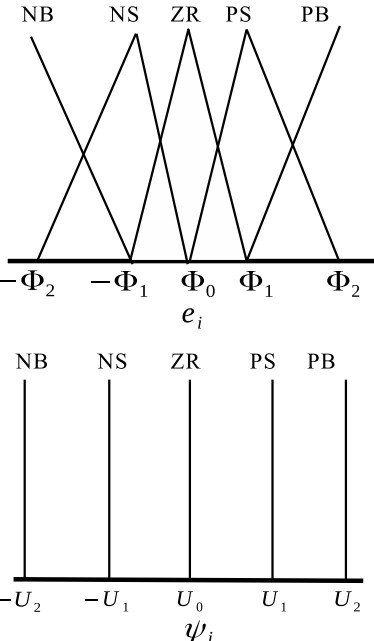

**Figure 2.** Membership functions for the input $e_i$ and the output $\psi_i$.

Meanwhile, the fuzzy system satisfies the following properties [22,26]:

(i) $\psi(e_i)$ is continuous and bounded.
(ii) $\psi(0) = 0$.
(iii) $\psi(e_i) = -\psi(-e_i)$.

The right-hand side of $\psi$ is represented as follows:

$$\psi = \sum_i \left\{ \frac{M_i}{M_r} \right\} U_i.\tag{15}$$

Here,

$$M_r := M_i + M_{i+1}$$
$$= \frac{e_i - \Phi_{i+1}}{\Phi_i - \Phi_{i+1}} + \frac{e_i - \Phi_i}{\Phi_{i+1} - \Phi_i}$$
$$= 1$$

because only two rules are fired at the same time. The fuzzy inference system (15) is calculated as follows [22,26]:

$$\psi(e_i) = \frac{M_i}{M_r} U_i + \frac{M_{i+1}}{M_r} U_{i+1}\tag{16}$$

for $\Phi_i \leq e_i < \Phi_{i+1}$; thus,

$$\psi = M_i U_i + M_{i+1} U_{i+1}$$
$$= \frac{\Delta U}{\Delta \Phi} e_i + \frac{1}{\Delta \Phi} \left( \Phi_{i+1} U_i - \Phi_i U_{i+1} \right)'\tag{17}$$

where $\Delta U := U_{i+1} - U_i$, and $\Delta \Phi := \Phi_{i+1} - \Phi_i$. Therefore, it is concluded that the control output $\psi$ is proportional to the input signal $e_i$ as follows:

$$\psi \propto e_i.$$

## 4. Results

In this section, numerical simulations obtained by the system formed by system (1), as well as the proposed observers of system (4) and (12), respectively, under a master-slave configuration are presented. For numerical simulation, the implicit Euler method with step size $h = 1 \times 10^{-3}$ is used to solve the systems (1), (4) and (12) [27]. The parameter values considered of system (1) are the same introduced in system (2). The initial conditions for hyperchaotic system (1) are selected as $x_1(0) = 20$, $x_2(0) = 100$, $x_3(0) = 600$, $x_4(0) = 0$ and for both observers are $\hat{x}_1(0) = 50$, $\hat{x}_2(0) = 200$, $\hat{x}_3(0) = 400$, $\hat{x}_4(0) = 0.5$.

Two cases are studied: first, the convergence of estimated variables given by SMO is tested. Later on, the convergence of estimated variables given by FSMO is considered. The both cases aforementioned were tested considering $F_1^+ = 40{,}000$ and $F_1^+ = 20{,}000$, thereby the observer gains were selected as $L_1 = 5 \times 10^4$ and $L_2 = 3 \times 10^4$. In Figure 3, the convergence of estimations given by SMO is illustrated.

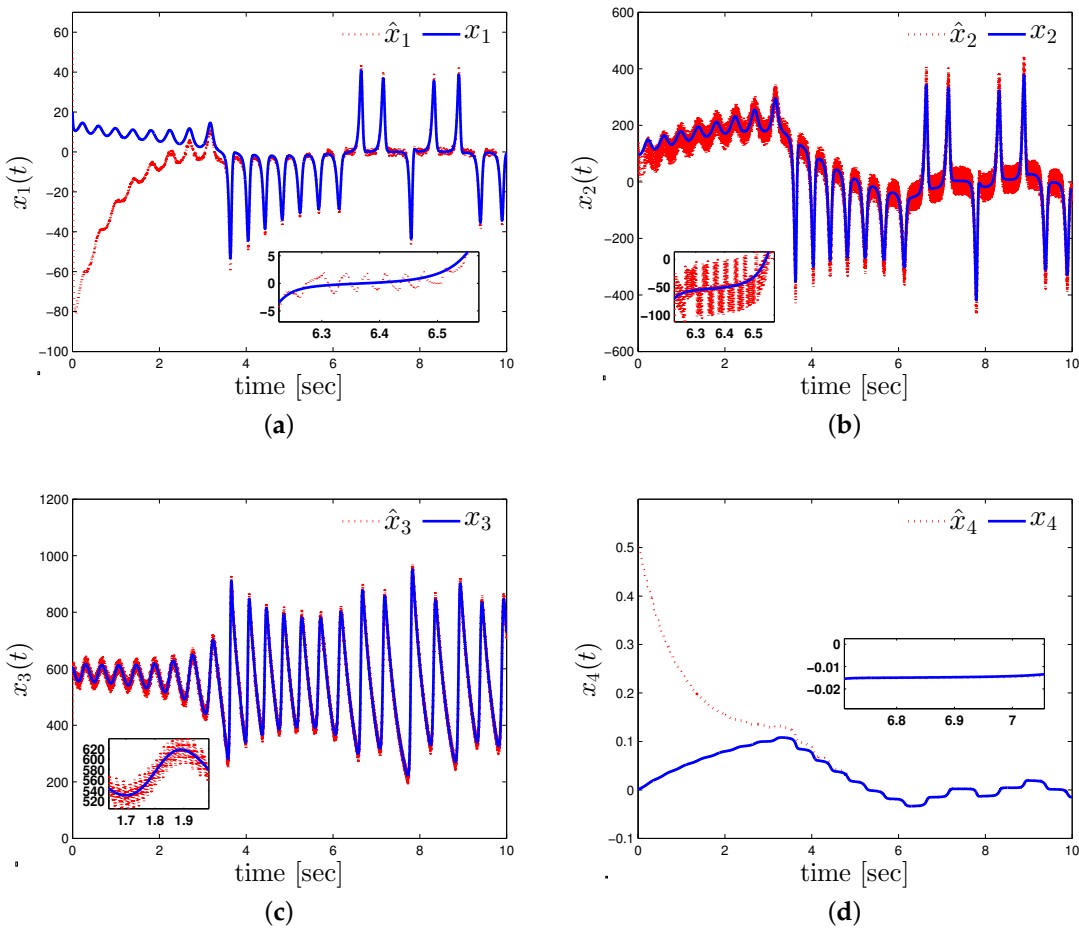

**Figure 3.** Actual (solid line) and estimated (doted line) values: (**a**) Dynamic of state $x_1$ and its estimated $\hat{x}_1$. (**b**) Dynamic of state $x_2$ and its estimated $\hat{x}_2$. (**c**) Dynamic of state $x_3$ and its estimated $\hat{x}_3$. (**d**) Dynamic of state $x_4$ and its estimated $\hat{x}_4$.

For FSMO, the final partition of universe of discourse are $e_2, e_3 \in \{-200, -20, 0, 20, 200\}$ and $\psi_2, \psi_3 \in \{-1, -0.7, 0, 0.7, 1\}$. The trial-error approach is used to tune the fuzzy system to obtain the best performance of FSMO. The convergence of the estimation provided by FSMO is depicted in Figure 4.

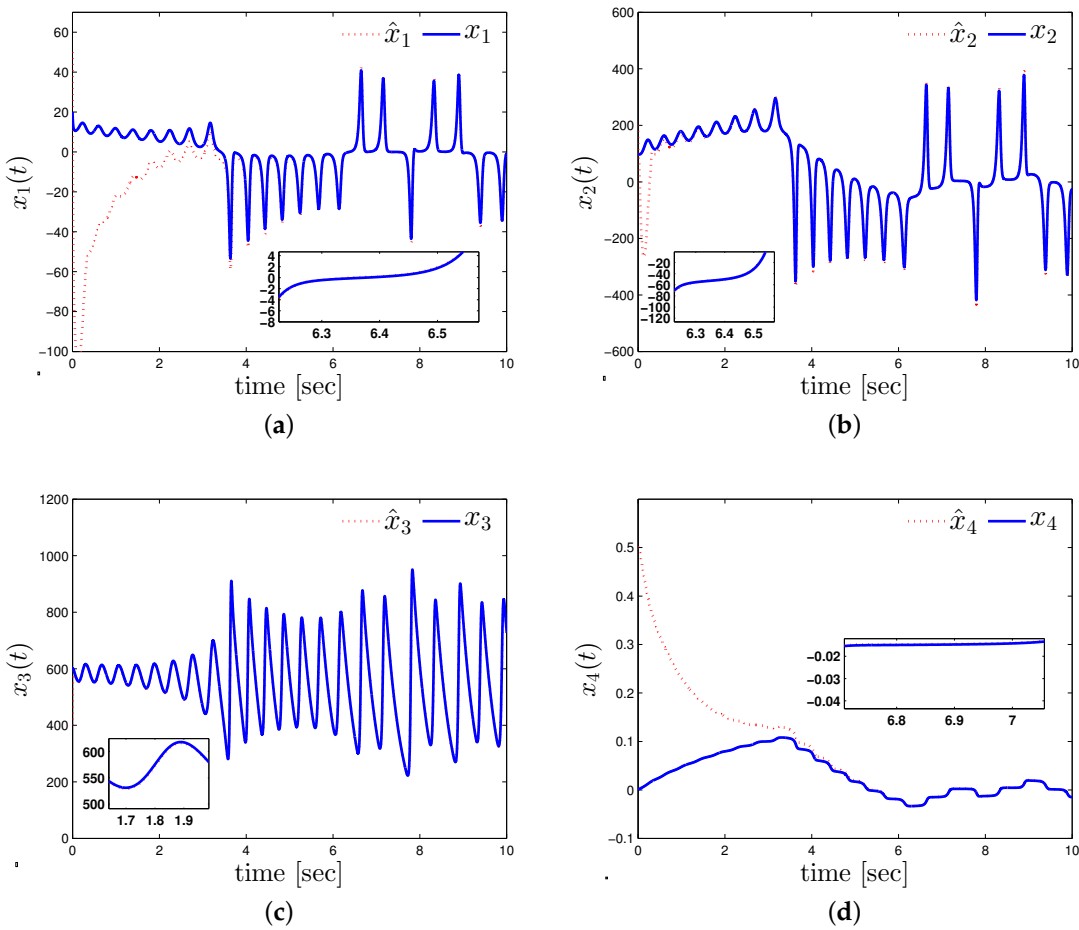

**Figure 4.** Actual (solid line) and estimated (doted line) values: (**a**) Dynamic of state $x_1$ and its estimated $\hat{x}_1$. (**b**) Dynamic of state $x_2$ and its estimated $\hat{x}_2$. (**c**) Dynamic of state $x_3$ and its estimated $\hat{x}_3$. (**d**) Dynamic of state $x_4$ and its estimated $\hat{x}_4$.

To evaluate the performance of both observers, the Mean Absolute Error (MAE) and Integral Absolute Error (IAE) are applied to the SMO and FSMO results. As can be seen in Table 1, the performance of FSMO is more accurate than that of SMO due to the presence of chattering phenomenon. The above can be verified in Figure 5, where the estimation errors of SMO and FSMO are illustrated.

**Table 1.** Error criteria for sliding-mode observer (SMO) and fuzzy sliding-mode observer (FSMO). MAE = Mean Absolute Error; IAE = Integral Absolute Error.

| Error | MAE | | IAE | |
|---|---|---|---|---|
| | SMO | FSMO | SMO | FSMO |
| $e_1$ | 2.0727 | 1.2836 | $2.0726 \times 10^5$ | $1.2835 \times 10^5$ |
| $e_2$ | 25.923 | 3.2871 | $2.5923 \times 10^6$ | $3.2866 \times 10^5$ |
| $e_3$ | 15.2191 | 1.6070 | $1.5218 \times 10^6$ | $1.606 \times 10^5$ |
| $e_4$ | 0.0057 | 0.0053 | 571.45 | 532.32 |

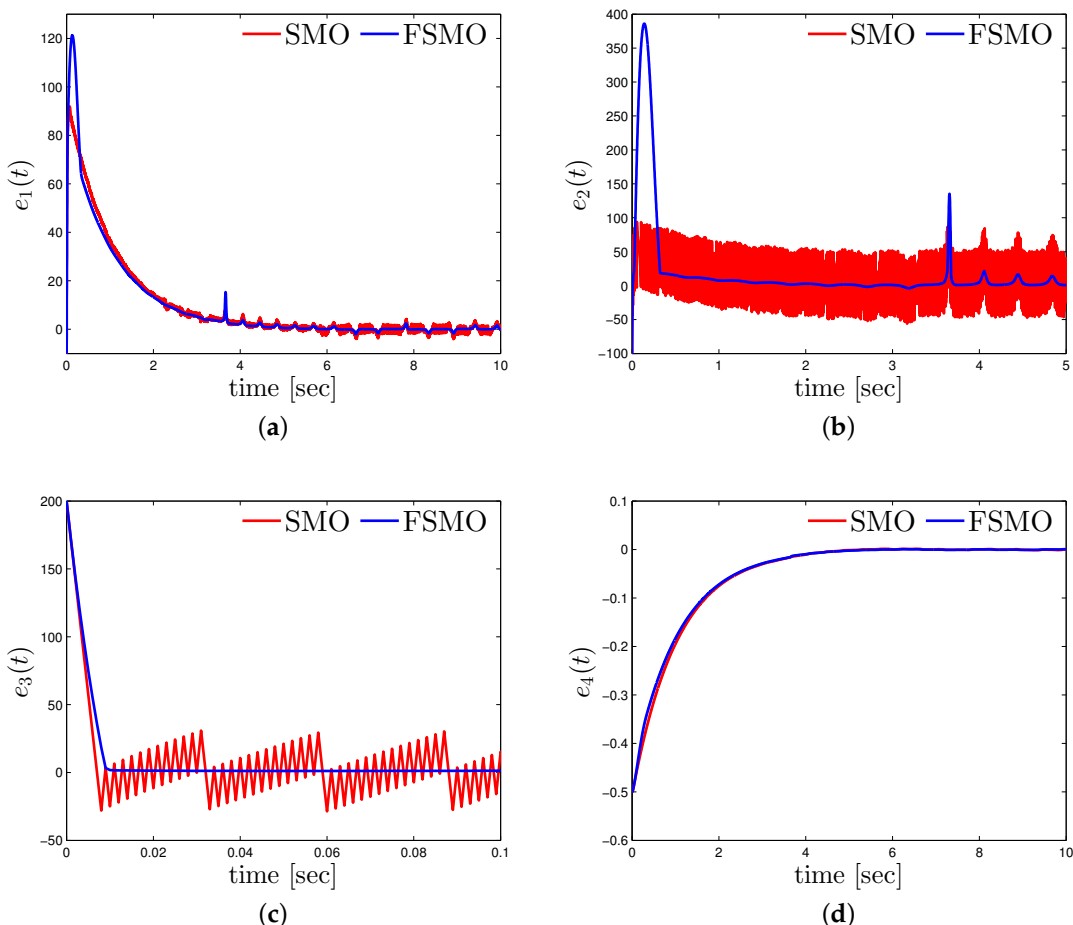

**Figure 5.** Time evolution of synchronization: (**a**) Error signals $e_1(t)$. (**b**) Error signals $e_2(t)$. (**c**) Error signals $e_3(t)$. (**d**) Error signals $e_4(t)$.

Since system (1) exhibits various attractors depending on the value of parameter $r$, the convergence of both observers is verified (see Figure 6) when the system (1) presents a periodic behavior. For the aforementioned, the following parameter values were considered:

$$b = 8/3,\ c = 1 + (159/200)\pi^2,\ r = 602.3,\ q = 1573, \alpha = 4/(9\pi^2),\ \sigma = 10,\ \zeta = 0.1. \tag{18}$$

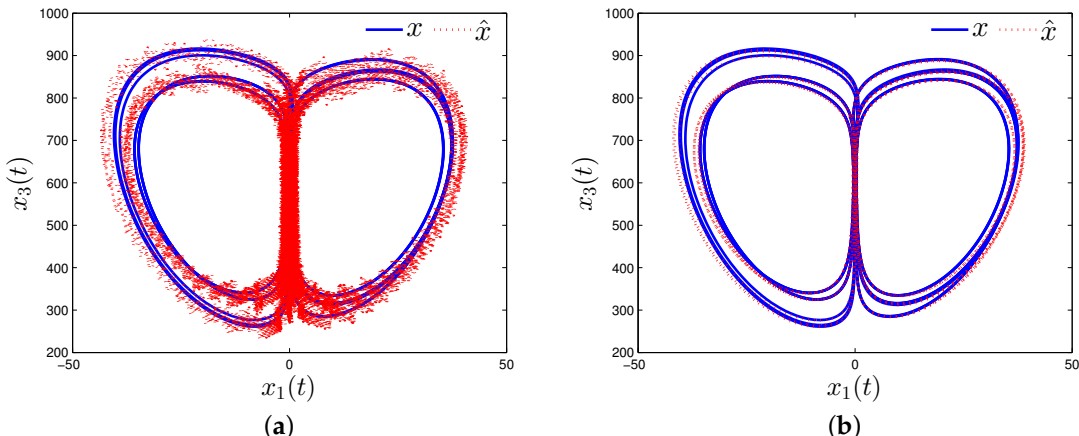

**Figure 6.** Phase portrait $(x_1, x_3)$: (**a**) SMO. (**b**) FSMO.

On basis of the results obtained in Figure 3, we can observe that the estimation variables exhibit chattering under SMO. Those high-frequency oscillations are introduced by the discontinuous implementation of the set-valued function. In contrast, we can see the absence of chattering in the estimation provided by FSMO.

## 5. Conclusions

We presented an FSMO and its application to solve the chaos synchronization problem of a hyperchaotic Lorenz-type system. First, we designed a SMO using the Lyapunov stability theory. The proposed SMO requires the measurement of two state variables from the hyperchaotic system. Later, we designed an FSMO by means of fuzzy inference in order to avoid the chattering problem derived of the SMO. More specifically, a fuzzy inference system was designed to replace the set-valued function $\mathrm{sgn}(\cdot)$. Finally, the effectiveness of synchronization for hyperchaotic system and the proposed observers were verified by the performance index MAE and IAE.

**Author Contributions:** Investigation, C.P., L.N.C., P.J.P. and R.R.-V.; Methodology, P.J.P. and R.R.-V.; writing—original draft preparation, C.P., P.J.P. and R.R.-V.; writing—review and editing, C.P., L.N.C., P.J.P. and R.R.-V. All authors have read and agreed to the published version of the manuscript.

**Funding:** This research was funded by TecNM grant number 6006.19-P "Observadores robustos para la solución de problemas de ingeniería".

**Conflicts of Interest:** The authors declare no conflict of interest. The funders had no role in the design of the study; in the collection, analyses, or interpretation of data; in the writing of the manuscript, or in the decision to publish the results.

## Abbreviations

The following abbreviations are used in this manuscript:

| | |
|---|---|
| ODE | Ordinary Differential Equations |
| FSMC | Fuzzy-based sliding-mode control |
| FSMO | Fuzzy-based sliding-mode observer |
| IAE | Integral Absolute Error |
| MAE | Mean Absolute Error |
| SMO | Sliding-Mode Observer |

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
