# Peer review of "Chaos Synchronization for Hyperchaotic Lorenz-Type System via Fuzzy-Based Sliding-Mode Observer"

_mca, doi:10.3390/mca25010016_

Round 1

Reviewer 1 Report

This paper presents a sliding mode based observer that enables a slave system to follow the master system where hyperchaotic dynamical patterns may occur. The introduction of the sliding mode observer is claimed to be able to eliminate the chattering phenomenon and thus improves the tracking performance of the slave system. The logic of this article is well designed and the writing is concise. Before the article can be accepted for publication, there are a few comments that need to be addressed: 1. To overcome chattering in sliding mode observer/controller, there are also other popular ways, such as setting a boundary layer around the sliding mode or simply utilizing a low pass filter to mimic smoother switching function. Thus, it is suggested that the author compare these different approaches of chattering elimination. Also, it is necessary to highlight the potential advantages of fuzzy base SMO as compared with its counterparts. 2. The boundness of nonlinear terms as stated in A1 should have their values shown in the numerical case studies since this will guide the selection of observers gain L1 and L2. 3. It is suggested that the author clarify the usage of the Euler integration scheme. The standard nonlinear dynamical study usually uses the 4th order RK algorithm is to ensure higher-order accuracy in time-domain simulation. 4. Meanwhile, the step size of 0.001 implies that the simulation is only capable of grabbing the dynamics below 500 Hz range. Note that \hat{x_2} shown in Figure 3(b) depicts high-frequency chattering due to the hard switching term in SMO, the choice of 0.001 step size needs to be validated to match the physics. 5. It is aware to the reviewer that proper tuning of the fuzzy logic parameter is key to ensure faster convergence and better tracking performance, as described at line 88-91. However, it should also be noticed that the error dynamics may be greatly influenced by the observer gain. Large observer gain may oversteer of the error around the equilibrium. Interestingly, the gain is also correlated to the prior knowledge of the magnitude of nonlinear terms. Thus, it is suggested that the author further explain the logical relationships between the bound F_1/2 as the set of optimal fuzzy parameters. 6. Since the presented system is hyperchaotic and the steady-state may depict various patterns as noted below Equation (1), it is suggested that tracking performance of the slave system under different dynamical patterns being evaluated to strengthen the superiority of SMO. In other words, different initial conditions of the master system that lead to different attractors need to be examined.

Reviewer 2 Report

Synchronization of chaotic dynamic systems has been a popular topic. This paper added a small twist to the synchronization strategy, i.e. the fuzzy sliding mode control. The paper is weakly acceptable. Some points must be clarified before it can be accepted.

Assumption 1 makes the problem super easy to handle. The authors may want to justify the reason for such a simplifying assumption.

In Eq. (4b) and (4c), x1*x2 and x1*x3 are used. Note that x1 is not available according to authors' assumption of the outputs x2 and x3. Hence, x1_hat should be used.

In Eq. (6b) and (6c), the terms x1*x2 and x1*x3 are still there and unchanged. This is obviously wrong.

The same issues with these two terms continue later in the paper.

What is more important is the consequence of this potential error: the stability proof may no longer hold.

Round 2

Reviewer 2 Report

I am happy with the revision.